# Recovery of Spinal Walking in Paraplegic Dogs Using Physiotherapy and Supportive Devices to Maintain the Standing Position

**DOI:** 10.3390/ani13081398

**Published:** 2023-04-19

**Authors:** Mădălina Elena Henea, Eusebiu Viorel Șindilar, Liviu Cătălin Burtan, Iuliana Mihai, Mariana Grecu, Alina Anton, Gheorghe Solcan

**Affiliations:** 1Phisiotherapy Unit, Clinics Department, Faculty of Veterinary Medicine, Iasi University of Life Sciences, Ion Ionescu de la Brad, 700490 Iasi, Romania; madalina.henea@yahoo.com; 2Surgery Unit, Clinics Department, Faculty of Veterinary Medicine, Iasi University of Life Sciences, Ion Ionescu de la Brad, 700490 Iasi, Romania; lburtan@uaiasi.ro (L.C.B.); iuliabogdan2005@yahoo.com (I.M.); 3Pharmacy Unit, Preclinics Department, Faculty of Veterinary Medicine, Iasi University of Life Sciences, Ion Ionescu de la Brad, 700490 Iasi, Romania; mgrecu@uaiasi.ro; 4Internal Medicine Unit, Clinics Department, Faculty of Veterinary Medicine, Iasi University of Life Sciences, Ion Ionescu de la Brad, 700490 Iasi, Romania; anton.alina@uaiasi.ro

**Keywords:** dog, spinal cord injury, paraplegia, physiotherapy, recovery of spinal walking

## Abstract

**Simple Summary:**

Spinal cord injuries (SCI) in dogs have become increasingly common and most are caused by trauma or discal hernia (intervertebral disc disease). The absence of deep pain perception is usually associated with very severe spinal cord injuries, manifested by paraplegia, urinary and fecal incontinence, decubital sores, and secondary infections. Due to the poor prognosis for recovery of voluntary locomotor functions, these patients are frequently considered for euthanasia. The main objective of our study was to demonstrate that physiotherapy and assisted gait in supportive devices to maintain the standing position may help paraplegic dogs to develop spinal walking. Of 60 paraplegic dogs without deep pain in the hindlimbs, 35 (58.33%) developed spinal walking.

**Abstract:**

Paraplegic patients have always been ideal candidates for physiotherapy due to their body’s inability to recover on its own. Regardless of the cause that led to the onset of paraplegia (traumatic or degenerative), physiotherapy helps these patients with devices and methods designed to restore the proper functioning of their motility, as well as their quality of life. A total of 60 paraplegic dogs without deep pain in the hindlimbs caused by intervertebral disc extrusion or thoracolumbar fractures underwent physiotherapy sessions: manual therapy (massage), electrostimulation (10–20 min with possible repetition on the same day), ultrasound therapy, laser therapy, hydrotherapy, and assisted gait in supportive devices or on treadmills to stimulate and relearn walking, which was the main focus of the study. To maintain the standing position over time, we developed different devices adapted for each patient depending on the degree of damage and the possible associated pathologies: harnesses, trolleys, straps, exercise rollers, balancing platforms and mattresses, physio balls and rollers for recovery of proprioception. The main objective of our study was to demonstrate that physiotherapy and assisted gait in supportive devices to maintain the standing position may help paraplegic dogs to develop spinal walking. Concurrent pathologies (skin wounds, urinary infections, etc.) were managed concomitantly. Recovery of SW was evaluated by progression in regaining the reflectivity, nociception, gait score, and quality of life. After 125 to 320 physiotherapy sessions (25 to 64 weeks), 35 dogs (58.33%) developed spinal walking and were able to walk without falling or falling only sometimes in the case of a quick look (gait score 11.6 ± 1.57, with 14 considered normal), with a lack of coordination between the thoracic and pelvic limbs or difficulties in turning, especially when changing direction, but with the recovery of the quadrupedal position in less than 30 s. The majority of dogs recovering SW were of small size, with a median weight of 6.83 kg (range: 1.5–15.7), mixed breed (n = 9; 25.71%), Teckel (n = 4; 11.43%), Bichon (n = 5; 14.28%), Pekingese (n = 4; 11.43%), and Caniche (n = 2; 5.71%), while those who did not recover SW were larger in size, 15.59 kg (range: 5.5–45.2), and mixed breed (n = 16; 64%).

## 1. Introduction

Spinal cord injuries (SCI) in dogs are common and most are caused by trauma or intervertebral disc extrusion (IVDE) [1]. The degree of recovery depends on the severity of the primary lesion, the percentage of the spinal cord affected, and the time elapsed from the time of injury to the time the patient presents for consultation. The pathways that carry deep pain perception (spinothalamic tracts) are more resistant to damage than other pathways, including those responsible for proprioception, motor function, and superficial pain [2], being located in the deepest areas of white matter [3]. The absence of deep pain perception is usually associated with very severe spinal cord injury [1,3], being considered evidence of functional spinal cord transection [3] and having a poor prognosis for the recovery of voluntary locomotor function [3,4].

Spinal walking (SW) is described as the acquisition of an involuntary motor function in paraplegic dogs and cats without pain perception affected by a thoracolumbar lesion [4]. SW is a reflex gait, resulting from complex dynamic interactions between the pelvic limb locomotor central pattern generator (CPG) and proprioceptive feedback from the body in the absence of superior control by the brain after complete spinal cord damage [5,6]. CPG is the network of interconnected interneurons in the spinal cord gray matter that modulates motor neuron activity for the generation of the gait [4].

The basic rhythmic pattern of the CPG is produced by interconnected, alternating, and mutually inhibitory flexor and extensor interneurons [5]. These interneurons, in turn, activate lower motor neurons via additional intermediary interneurons, the output from which serves as the final common pathway for producing locomotion via direct innervation of appendicular muscles [5]. The CPG also provides coordination between left and right limbs via the integration of commissural interneurons and thoracic and pelvic limbs, important in normal quadrupedal locomotion [6].

The central nervous system is widely considered to have poor regenerative capacity; however, remarkable plasticity is possible, being demonstrated on various spinal cord transections and decerebrate animal models [6,7,8,9,10]. Reorganization and adaptations that might influence the recovery of motor function below the level of severe injury include regrowth of axons across the epicenter, recovery/reactivation of conduction of residually intact upper motor neuron (UMN) axons traversing the lesion epicenter, a more autonomous role for the CPG, alterations in excitability of interneurons and LMNs below the injury, activation of silent synapses, changes in synaptic weight, and alterations in sensory input or how afferent input is integrated at the level of the spinal cord below injury [5,6]. The inability of axons to regenerate after a spinal cord injury in the adult mammalian central nervous system (CNS) can lead to permanent paralysis [11,12].

Many attempts have been made to recover paralyzed pets and to improve their quality of life, including surgical interventions (hemilaminectomy) [7,13,14,15,16,17,18,19,20], antiinflammatory (prednisone, dexamethasone, methylprednisolone sodium succinate, meloxicam, firocoxib, and celecoxib) and nutraceutical treatments (vitamin B1, B6, E, glucosamine, and chondroitin) [3,13,14,15,16], acupuncture and electroacupuncture [21,22,23], decompressive surgery followed by electroacupuncture [24], physiotherapy (physical therapy, hydrotherapy, treadmill, laser therapy, ultrasounds, and electric stimulation,) [7,13,25,26,27,28,29,30], cell-based (regenerative) therapy [3] or combinations of different techniques, as they are frequently considered for euthanasia [17,31]. According to the newest consensus statements of the American College of Veterinary Internal Medicine (ACVIM) on the management of acute canine thoracolumbar intervertebral disc extrusion (IVDE), surgical management might be considered in a young, active dog with multiple mineralized discs, particularly with recurrent events [32]. Surgical decompression can be considered when neurologic signs are progressive or unimproved or if the pain is persistent despite appropriate medical management [32]. Ambulatory dogs can be managed successfully medically; however, consideration should be given to the risk of recurrence [32].

In one study, SW was obtained after a period of intensive physical rehabilitation training, consisting of passive range of motion exercises, flexor reflex and crossed extensor reflex stimulation, active-assisted exercises, electrostimulation, and hydrotherapy on an underwater treadmill, which was preceded by surgical management in 59% of the patients without deep pain perception [7].

Physiotherapy includes all the physical factors involved with an organism, allowing for the development of a treatment plan for complete recovery, but it also has prophylactic purposes. It comprises manual therapy (massage), passive range of motion exercises, active-assisted exercises, hydrotherapy, treadmill or underwater treadmill therapy, ultrasound therapy, laser therapy, physical therapy (electrostimulation, light, sound, heat, and cold applications), and movements [25,26,27,28,29,30,33]. Manual methods aim to restore muscle mass, strength, balance, and physiological movements. Physiotherapy aims to help the body reach a critical threshold by restoring quality of life [34,35]. Locomotor, neurological, and orthopedic diseases are the target towards which physiotherapy is oriented, encouraging the body to regain its functionality. Techniques for obtaining and maintaining the standing position are used for patients who have lost the ability to maintain an anatomical standing position, making it impossible for them to support their own weight. Physiotherapy is a combination of therapeutic methods and techniques that help the body recover when it is not capable of doing so itself [25,26,27,28,29,30]. Different scoring systems have been developed over time to evaluate the degree of recuperation of motility and gait [1,18,36,37]. One of the key points regarding the application and importance of improving current physiotherapy for dogs with SCI is the development of chronic pain [14,29,32,38].

Taking into account that the absence of deep pain perception is usually associated with very severe spinal cord injury, being considered evidence of functional spinal cord transection, the main hypothesis of our study was that some paraplegic dogs may recuperate spinal walking using physiotherapy techniques and devices adapted for the restoration and maintenance of the anatomical standing position.

## 2. Materials and Methods

The study was carried out at the Faculty of Veterinary Medicine in Iași at the physiotherapy service department on 60 paraplegic dogs without deep pain in the hindlimbs and no proprioception caused by IVDE or thoracolumbar fractures located between the Th3 and L3 vertebrae (Appendix A). The absence of pain perception was assessed as a lack of a conscious response (e.g., crying, looking around, or a similar reaction) to the application of heavy pressure to the pelvic limb digits with forceps. The sites and types of lesions are presented in Table 1.

Each dog was evaluated clinically and then neurologically according to a scoring system recommended by Olby et al. [37], and the patients were neurologically re-evaluated every 10 sessions. Dogs were included in the study if they had a medical record documenting paraplegia and absent pelvic limb pain perception after physical and neurological examinations by the referring neurologist and at admission to the physiotherapy unit; a spinal cord lesion was confirmed by radiology and in some cases myelography, computed tomography, or magnetic resonance imaging (MRI) and the absence of pelvic limb pain perception at admission according to the physiotherapy rehabilitation treatment protocol. Dogs that developed myelomalacia were excluded. Initially, the animals were allowed to adapt and get used to the equipment and the environment, as well as the staff that they would come into contact with during the therapeutic sessions.

Depending on the type of lesion and the needs of each individual patient, an individual physiotherapy protocol was established. The physiotherapy sessions sequence order was as follows: manual therapy (massage), electrostimulation (10–20 min with possible repetition on the same day, Text S1), ultrasound therapy, laser therapy, active and passive range of motion, and assisted gait in supportive devices or on treadmills to stimulate and relearn walking. Laser and ultrasound therapy was administered using the Intelect Vet Therapy System (Orto Canis), massage, and active/passive movement exercises.

Electrostimulation was carried out daily on the muscles of the affected pelvic limbs, starting with 15 min of treatment and increasing the duration of electrostimulation during the sessions up to 40 min per session using the following parameters:-For spasm reduction, the current type was Premod (interferential), the type of emission was continuous, low frequency (10 Hz), and intensity was variable.-For muscle strengthening, the current type was VMS (vibratory motor stimulation), with continuous emission, phase duration of 200 μs, a ramp of 2 s, frequency of 50 Hz, and variable intensity (Appendix A).

Transcutaneous electrical neuromuscular stimulation (TENS) was used to manage short- and long-term pain in physical therapy. The method implies the application of electrodes on areas of the body where the pain is present and inducing an electrical current of intensity and frequency that would interfere with the normal pain signals emitted by the sensitive cells in the affected area, thus reducing the pain sensation perceived by the patients [25].

Neuromuscular electrical stimulation (NMES) aimed to mimic a physiological motor neuron impulse, and thus stimulate an isolated muscle or group of muscles with the express intention of countering muscle atrophy. Thus, by artificially promoting a contraction that leads to the usage of a muscle’s reserves, we stimulate the regeneration of the tissue and its hypertrophy back to its normal, physiologically functional level. NMES was also used as a curative therapy, the muscle movements created via electrical currents promoting blood flow, acting as a tissular decongestant. Furthermore, NMES was used in the treatment of muscular spasms, by prolonged exposure to electrical currents to ”tire out” the muscle of its reserves, thus helping it relax by breaking the stimulation–contraction feedback loop [25,39].

Interferential current (IFC) was used in the treatment of pain, to relieve muscle spasms, and to improve blood flow in various tissues, using the principles of NMES stimulation. The technique uses multiple electrodes, typically four, placed in a crisscross position. This arrangement causes the currents running between the electrodes to ”interfere” with one another, thus allowing the therapist to use a higher intensity current whilst still maintaining a maximum level of comfort for the patient [25].

Motor points for electrostimulation were selected according to Goody [40] (Appendix A).

Each dog in the study received daily physiotherapy sessions (from Monday to Friday) lasting approximately one hour per session. Treadmill exercises were the most commonly used procedures in this study. After the patients become accustomed to the harness and their muscular tissue developed, they were introduced to the treadmill. In the beginning, low speed was used in order to stimulate the animal to make physiological movements. After a few sessions, the speed increased along with the patients’ progress. At first, a speed of 0.8 km/h for 15 min was applied for the first 40 sessions, and afterward, between the 40th and the 80th session, the treadmill was set to a speed of 1.5–1.8 km/h for 20 min. After this, up to the 125th session, 2.5 km/h for 30–45 min was used. In patients that required more than 125 sessions, 2.8–3 km/h speed for 1 h was applied. The SW recuperation score was evaluated weekly when walking on the treadmill [37] by two independent evaluators (MEH and GS).

The duration of the physiotherapy program was adapted according to each individual patient, taking into account their diagnosis and prognosis. The duration of the physiotherapy treatment cycle was also adapted to each individual patient based on the owners’ decision on whether to continue physiotherapy sessions.

After conducting a thorough clinical examination of each patient, treatment protocols were established using the methods and devices developed over time. At the same time, the exercises were chosen so that the patient was completely safe and would achieve a satisfactory recovery. During therapeutic sessions, signs of improvement in the patient’s health indicate the need to change the treatment plan so that the intensity, duration, and frequency of an exercise will be increased or decreased depending on the patient’s condition [28,29,30]. As for the materials, they were chosen precisely in terms of their accessibility. The main materials used to obtain the devices meant to restore the standing position were metal frames with a pulley, stable metal frames, wheeled metal frames, scarves, and towels. The devices were adapted from classical literature on rehabilitation protocols [26,27,28,29,30].

1. Manual methods of therapeutic exercises involved common activities such as walking on a treadmill, stair climbing, and dancing (to stimulate the hindlimbs). In addition to these common activities, other techniques such as jogging, walking, playing with and moving toys, and crossing tunnels were implemented. Game balls were very useful.

2. Techniques for obtaining and maintaining the standing position.

### 2.1. Total Assistance for Supporting Body Weight and Adopting the Standing Position

This was applied to animals that have completely lost their ability to remain in the anatomical position, especially after paralysis (Figure 1). In this case, total assistance means the support of 75–100% of body weight (bw) by the therapist [25]. Depending on the patient’s height and weight, the therapist can lift them directly, manually, or using equipment such as specially adapted towels or harnesses. The devices we made specifically to provide more support and to obtain the anatomical standing position were represented by stable metal frames. These metal frames allow for the attachment of harnesses or towels for the correct positioning of patients. Once the anatomical standing position is established, the therapist induces limb movements. Thus, the animal is helped to support its body weight with its feet on the ground.

### 2.2. Standing Position Actively Supported Using Trolleys and Straps

Trolleys and straps are very useful in physiotherapy sessions. In this regard, in order to obtain a trolley accessible to anyone, we have developed a wheeled metal frame to which a pulley has been added to attach the harness in which the animal is positioned. Thus, the pulley fixed at the top of the metal frame allows for the adjustment of the height at which the harness is located and implicitly induces the standing position of the animal.

In our clinic, the two variants of trolleys (with two or four wheels) are adapted according to the patient’s condition: those with two wheels are used for those with neuromotor disorders of the hindlimbs, while those with four wheels are used for patients in whom all four limbs are affected. In this study, only patients with T3–L3 injuries were included, so the first variant of the trolley was used. At the same time, the straps were useful for non-ambulatory patients (those who are unable to move). For the placement of the straps, the height of the animal was taken into account for the positioning of the limbs (performed manually by the therapist or automatically and physiologically by the animal) in a natural posture to support the body weight (Video S1).

### 2.3. Standing Position Actively Supported with Exercise Rollers

The active support of the animal was also achieved with an exercise roller of appropriate size (Figure 2). However, this method requires the presence of two people, especially in cases of medium or large patients. Once the size of the roller is adjusted, the animal is placed in a standing position on it so that all four limbs touch the ground. This method is more effective than using a physio ball considering its lack of stability. It is also recommended that incompletely inflated rollers be used instead of hard, fully inflated ones. Once the animal is placed on the roller, one person must provide support for the forelimbs, and the second person controls the hindlimbs. In this position, the therapist can induce up-and-down or swaying movements. At the same time, as the patient develops endurance and muscular strength, the speed with which this exercise is performed can be increased for better stimulation of balance and muscles.

### 2.4. Minimal Support of the Standing Position 

This is the next stage of the therapeutic process (Figure 3). Patients able to support themselves in the anatomical position with the necessary strength and control were included in this stage. However, in some cases, there is still weakness or incoordination, which can lead to a loss of balance, consequently requiring assistance to regain the standing position. The therapist places themselves on the patient’s side, being prepared to support them in case they lose balance, intervening only if absolutely necessary. As the animal’s condition improves, it increasingly seeks to move.

### 2.5. Recovering of Proprioceptive Sensitivity

Once the patient acquires the ability to maintain the anatomical standing position without the intervention of the therapist, the next stage aims to regain the patient’s balance (Figure 4). The dynamic balance protocol was used when the patient was able to maintain balance during motion. The techniques were applied on dry, non-slippery surfaces, allowing for physiological locomotion in safe conditions with a low risk of falling.

### 2.6. Changing the Center of Gravity 

Changing the center of gravity can be achieved by encouraging the animal with a treat or even a ball (Figure 5). The animal will carefully watch the treat or the ball, making up-and-down movements, as well as from right to left. Initially, the movements have a small amplitude, becoming more and more difficult during the therapeutic sessions. The key element in this stage is represented by the movement made with the head, which causes the patient to change their center of gravity using balance, strength, and coordination.

### 2.7. The Balancing Platform 

A balancing platform was used for the forward and backward, sideways, diagonal, and 360° swinging movements of the animal (Figure 6). The fore- or hindlimbs were placed on the platform, while the opposite pair was on the ground (Figure 7). Ideally, one person should provide the patient with support, and a second person should perform light and slow rocking movements on the platform.

### 2.8. Physio Balls and Rollers

For the development of balance, coordination, and strength in animals, balls and exercise rollers intended for human use were used (Figure 8). These devices have proven to be useful even in cases of common stretches. The exercises involved positioning the forelegs on the ball with the support of the therapist, forcing the animal to maintain static balance on the hindlimbs. At the same time, the dynamic balance test was performed by inducing forward–backward and lateral movements on the device so that the hindlimbs were forced to maintain balance throughout these movements. When intervention was required on the neck and the anterior region of the body, the hindlimbs were positioned above the device, forcing the forelimbs to maintain balance and support the body weight both when standing and during movement.

### 2.9. Motility Exercises

The devices developed to facilitate the movement of patients were represented by scarves and towels that were attached to the aforementioned metal frames or were handled directly by the therapist. Initiating an effective workout was achieved with the help of straps, towels, harnesses, or even trolleys (Video S1). The therapist chose the right device for the patient after a prior assessment of their needs. Every patient was treated using the harness, and afterward, the devices were chosen according to the recovery progress. In the beginning, each patient stays in the harness attached to the stable metal frame, and after they gain mobility, they are placed in the wheeled device. After a few treatment sessions, when they are able to maintain a standing position on their own, the animals may be carried using a towel or a scarf. The animal must be motivated and encouraged to move, even very slowly, as independently as possible. The therapist can intervene if the animal needs its limbs to be placed in an anatomical position.

Long towels can prove to be very useful in supporting animals that need help to maintain their standing position or to move. Additionally, a backup device is represented by scarves. Both options are useful, easy to find, and affordable. For the support of the rear part of the body, the towel was positioned around the abdominal region, right in front of the hindlimbs. To ensure the optimal length for handling, the ends of the towel were sewn with nylon strips. Recovery of SW was evaluated by progression in regaining the reflectivity, nociception, gait score [37], and quality of life [35].

A *t*-test was used to compare the two groups: spinal walking and no spinal walking. The values of the motility score were calculated with a commercially available statistical data analysis program (MedCalc version 20.215). The assessment of data for normality was calculated by applying the D’Agostino–Pearson test. Data with normal distribution were expressed as mean and standard deviation (SD), while when normality was rejected, a median with a 95% confidence interval (95% CI) was used. Values of *p* < 0.05 were considered significant.

## 3. Results

All patients included in the study were diagnosed with paraplegia and urinary incontinence as a result of sudden onset thoracolumbar IVDE, spinal cord fracture, or luxation with no deep pain and no proprioception. The patients with IVDE from this study were presented to the clinic after the critical first 72 h after the injury. Hemilaminectomy was performed in 44 of 53 patients (83.01%), but, when considering the time frame, the surgery was performed too late, when the spinal cord was severely damaged. Additionally, the intervertebral disc extrusion was very large in some patients, which made the recovery process difficult or even impossible. All the patients with SC trauma (7/60; 11.66%) benefited from surgical stabilization.

Spinal walking (SW) was obtained progressively after 125–320 physiotherapy sessions, taking from 25 to 64 weeks, with an average of one year for most patients (Table 2 and Table 3). SW was defined as the ability to develop an unassisted involuntary gait [6,7]. Dogs were considered spinal walkers if they could walk for a potentially infinite period of time and were able to regain the standing posture and continue to walk unassisted in the case of a fall [7].

Statistically significant differences were observed only for weight, *p* = 0.0085. The no spinal walking group was constituted of dogs with a higher weight, which may suggest the fact that the chances of recovering spinal walking are greater in dogs with a lower weight.

Even if there was no reflectivity at the time of injury, some patients may recover it spontaneously or after physiotherapy. Patients with injuries to the T9–T10 (2/60), T10–T11 (6/60), and the L1–L2 (1/60) intervertebral spaces were presented at the clinic with no proprioception or withdrawal reflex. Additionally, from the 29/60 patients with injuries to the T13–L1, only 6 did not have a withdrawal reflex or proprioception. After 20 sessions, we observed that 18 patients (30%) regained reflectivity, an important step in the automatic development of SW. After 30 sessions, all of the patients regained deep pain and tail mobility. The other 23 patients (38.33%) with injuries at the T13–L1 were presented to the consultation with a withdrawal reflex. The patients with injuries at the T11–T12 vertebrae came to the consultation with withdrawal reflex present. However, of the 60 patients, 12 (20%) lost reflectivity definitively (Table 3). All these patients belonged to the group of 25 who did not develop SW.

After 125 to 320 physiotherapy sessions (25 to 64 weeks), 35 dogs (58.33%) developed SW and were able to walk without falling or falling sometimes in the case of a quick look (gait score 11.6 ± 1.57, with 14 considered normal), with a lack of coordination between the thoracic and pelvic limbs and difficulties in turning especially when changing direction but with the recovery of the quadrupedal position in less than 30 s. All patients who developed SW while lacking cortical coordination of the hindlimbs kept some specific gait dysfunctionalities: they could not jump with the hindlimbs, they could not jump into bed or on the couch, and when running away they could not jump with both hindlimbs (Video S2). Some patients who managed to go up and down the stairs did not use their hindlimbs alternately but made their pelvis thrust and put both limbs at once on the next step because they could not jump. The site of the lesion was not significantly associated with the degree or time of recuperation of SW. A Pearson correlation test showed statistically significant positive associations (r > 0.5) between gait score and the number of physiotherapy sessions. The strongest association (r = 0.9) was observed between t0 and t3, indicating that a higher number of physiotherapy sessions results in a better spinal walking score (Table 3).

The other 25 dogs (41.67%) remained with permanent neurological deficits. Two of them (8%) died due to relapsing urinary infections that were very resistant to antibiotic therapy. The others were helped to move with veterinary prostheses or various devices (harnesses) used to maintain the standing position or were euthanized at the behest of the owners.

## 4. Discussion

The main hypothesis of our study was that some paraplegic dogs may recuperate spinal walking using physiotherapy techniques and devices adapted for the restoration and maintenance of the anatomical standing position. The permanent lack of pain perception has been commonly, and frequently incorrectly, interpreted as an indication of spinal cord transection, complete disconnection from all supraspinal influence, and minimal to no chance of meaningful recovery of function [6]. However, a proportion of permanently deep-pain-negative dogs demonstrate notable spontaneous motor recovery over time [1,7,41,42]. After 125 to 320 physiotherapy sessions, consisting of manual methods, physical therapy (electrostimulation, laser, and ultrasound), techniques for obtaining and maintaining the standing position, assistance in supporting body weight and adopting the standing position, electrostimulation (Appendix A), techniques for recovering of proprioception and motility exercises, 35 of 60 paraplegic dogs (58.33%) developed SW and were able to walk without falling, or falling sometimes in the case of a quick look (gait score 11.6 ± 1.57, with 13 considered normal). Our results are quite similar to those of Gallucci et al. [7], who obtained SW in 59% of 81 paraplegic dogs without pain perception using five basic categories of exercises differently arranged: passive range of motion exercises, flexor reflex and crossed extensor reflex stimulation, active-assisted exercises, electrostimulation, and hydrotherapy on an underwater treadmill. Yu et al. [43] showed that activities directed toward the restoration of movement, such as the use of “over-ground” treadmill training (as shown in people to improve motor recovery and balance (e.g., using proprioceptive platforms) may actually be more rational early on in the recovery process for paraplegia induced by intervertebral disc extrusion (IVDE) and should be encouraged first, but an “under-water” treadmill is best suited to improve strength and muscle mass. In human beings, assisted walking movement in patients with genetic and acquired neuromuscular disorders with a motorized Innowalk device gave better results [44]. Using an intensive neurorehabilitation (INR) training protocol based on the evidence of signal transmission throughout the lesion caudally and rostrally, which can be detected by electromyography, and a multimodal approach, which included locomotor training, electrostimulation protocols, and, in circumstances of absent deep pain perception, pharmacological management, in dogs previously treated with hemilaminectomy for IVDE, Martins et al. [45] obtained an ambulation rate of 58.5% within a maximum of 3 months.

A retrospective study of physiotherapy conducted by Jeong et al. [46] evaluated the neurological outcomes in dogs with IVDE causing injuries ranging from ambulatory paraparesis to paraplegia with absent nociception. The authors reported a significant improvement in locomotor outcomes for dogs receiving surgical decompression coupled with physiotherapy when compared to those receiving surgical decompression alone. However, successful locomotor outcomes for dogs with paraplegia with or without intact deep pain receiving decompressive surgery alone were only reported to be 17%, which is much lower than the typical outcomes reported in the literature, which range from 50 to 60% [13].

In a study that assessed long-term outcomes in paraplegic dogs without deep pain perception on the pelvic limb [1], 64/70 (58%) developed SW after surgery for IVDE, but only 2/9 (22.2%) of those with spinal cord trauma developed SW.

There is no standardized physiotherapy program followed across the board, making it difficult to compare results between studies. The type of physiotherapy shown to improve locomotor outcomes in experimental injury models, and employed in people with severe injuries, is a very intensive type of locomotor training [44]. The therapeutic activities implemented in veterinary medicine classically include treadmill walking, passive range of motion exercises, and assisted weight-supported walking. The physiotherapy protocol must be established based on several factors, and the therapeutic program is developed precisely for the needs of each patient. The treatment program is then expanded after a clinical examination identifies the present conditions, techniques, and exercises that play a major role in effective therapy.

In our study, apart from classical physiotherapy tools such as electrostimulation, laser, ultrasound, and treadmill exercises used by physicians in clinics, we implemented different methods and tools aiming to obtain and maintain the standing position, which can easily be used by the owners at home. The purposes of maintaining the standing position are to reconsolidate muscle mass and endurance, regain proprioception, improve respiratory function and blood circulation, speed up the healing of decubitus ulcers, and support the patient’s psychological state. Orthopedic and neurological disorders are the main types targeted by these exercises [25]. Trolleys allow animals a sense of freedom, significantly reducing the support provided by the therapist. The animal should not be left hanging above the ground as it may affect the local blood supply and press areas directly in contact with the support devices attached to the trolley [25].

Playing with the ball, a more demanding activity was carried out by throwing the ball a short distance, up, and to the side. Initially, as a result of this exercise, the animal may show excessive enthusiasm that can lead to falls or bumps, requiring the presence of a person who is prepared to assist the animal. The therapist can test the animal’s ability to regain balance by gently pushing the shoulders or hips or by removing the healthy limb from the support. A change in the center of gravity can also be induced while strolling. While walking in a straight line, the therapist will gently push the animal to one side to encourage it to maintain its balance [26].

The balancing platform can be used easily and with very good results in animals, with the aim of developing proprioception [25]. The great advantage of motility exercises using towels and scarves is that they are cheap and available to anyone. However, a disadvantage of using towels in large animals with urinary incontinence is that the towel, because of its location, will put pressure on the bladder, causing urine to be excreted. However, this can easily be avoided by using a harness. Another disadvantage is that some towels may be uncomfortable for the therapist [25].

In a study by Gallucci et al. [7], the median duration between the start of physiotherapy treatment and the ability to stand up unassisted, recorded in 29 dogs, was 20 days (range 2–150). In our study, 47 dogs (78.33%) could stand up fourfold when drinking water or eating and could remain in this position for about 30 s to one minute after a median duration of 40 days, and 35/60 (58.33%) developed spinal walking after 125–320 physiotherapy sessions (25–64 weeks) (Video S2). The incomplete recovery of forelimb–hindlimb coordination in dogs with excellent recovery of stepping was clearly documented by the kinematic evaluation of the gait on a treadmill [46]. Since the prevalence of naturally occurring traumatic SCI is high within the general population of pet dogs, making this population of dogs a potential tool through which to screen new interventions for people with SCI to generate relevant translational data for humans [47], further studies should be encouraged to demonstrate our belief that physiotherapy could play a crucial role in the development of SW.

In our study, the site of the lesion was not significantly associated with the degree or time of recuperation of SW. Blauch [41] stated that SW could develop only if the lesion was cranial to the 13th thoracic vertebra, to avoid the atrophy of paravertebral muscles. Gallucci et al. (2017) achieved SW in 5 of 81 paraplegic dogs (6.17%) with spinal cord lesions at the presumed level of the CPG (L2–L3). Zidan et al. [38] demonstrated the positive effect of electromagnetic fields associated with some rehabilitation protocols on postoperative pain and locomotor recovery in dogs with acute, severe thoracolumbar IVDE without pain perception.

According to the newest consensus statements of the ACVIM on the diagnosis and management of acute canine thoracolumbar IVDE, basic rehabilitation exercises are recommended as an additional treatment (e.g., passive range of motion exercises and massage), with an emphasis on restricted activity for at least 4 weeks, followed by increased levels of physical activity [32].

There is currently limited evidence for specific treatments to facilitate the recovery of SW in dogs lacking pain perception. A variety of therapeutic interventions have been investigated in experimental models and human SCI to optimize recovery [6]. These include task-specific physical rehabilitation [48], functional electrical stimulation [49] and epidural stimulation [50], targeted somatosensory stimulation, treating neuropathic pain, and other pharmacologic interventions [11,45,50,51,52,53,54,55,56].

At the site of spinal cord injury, a glial scar develops containing extracellular matrix molecules including chondroitin sulfate proteoglycans (CSPGs), which are inhibitory to axon growth. Bradbury et al. [11], have demonstrated that intrathecal treatment with chondroitinase ABC (ChABC) degraded CS-GAG at the injury site, upregulated a regeneration-associated protein in injured neurons, and promoted regeneration of both ascending sensory projections and descending corticospinal tract axons. ChABC treatment also restored post-synaptic activity below the lesion after electrical stimulation of corticospinal neurons and promoted functional recovery of locomotor and proprioceptive behaviors. Combination therapy with task-specific training and chondroitinase ABC in experimental SCI models has been shown to promote regeneration and synergistic plasticity with a greater degree of effective synaptic connections reestablished below the injury in an activity-dependent manner [56,57]. There is growing evidence that multimodal approaches to facilitate motor recovery might prove most useful in improving outcomes in conjunction with traditional approaches directed at the lesion epicenter [5,58].

Our study has some limitations since for economic or technical reasons a limited number of cases have the benefit of modern imaging diagnostic tools (computed tomography and magnetic resonance) prior to accurate surgery. Ethical reasons did not allow us to have a control group of dogs, thus preventing the possibility of demonstrating the real influence of physiotherapy.

## 5. Conclusions

Our study demonstrates that the majority of paraplegic dogs without deep pain perception undergoing long-term physiotherapy treatment associated with the use of some simple, at-home devices to maintain the standing position (harnesses, trolleys, straps, exercise rollers, balancing platforms and mattresses, and physio balls and rollers for recovery of proprioception) followed by walking on a treadmill acquired SW gait. The chances of recovering spinal walking are greater in dogs with a lower weight. Regaining reflectivity is an important step in the automatic development of SW.

These findings may encourage the owners of paraplegic dogs to develop their own similar devices so that they can contribute to the therapeutic protocol and to the healing of patients.

## Figures and Tables

**Figure 1 animals-13-01398-f001:**
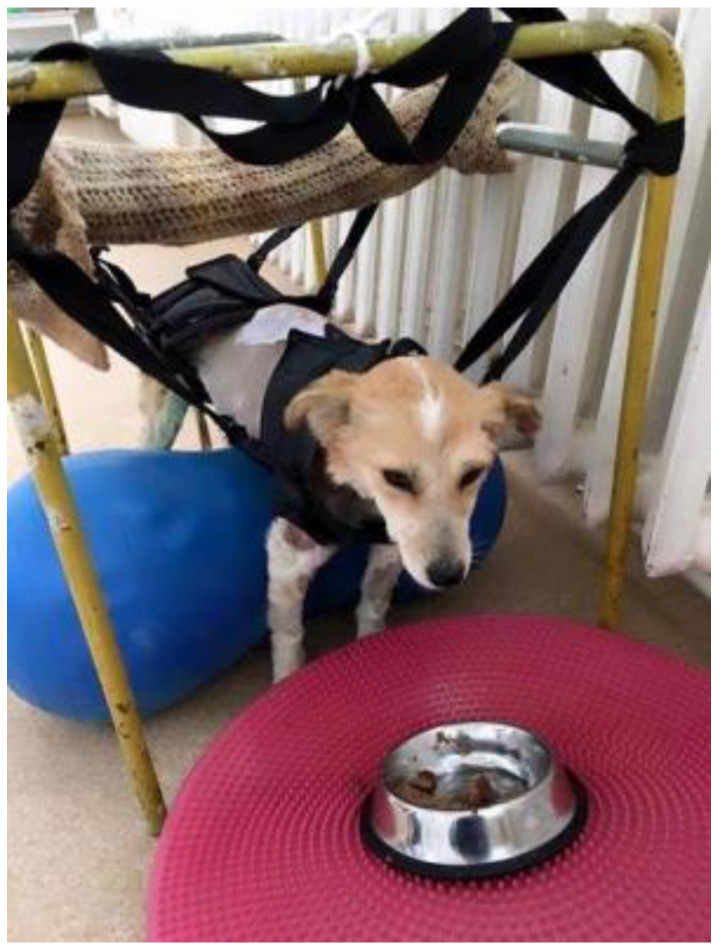
Paraplegic dog supported in the standing position by harnesses.

**Figure 2 animals-13-01398-f002:**
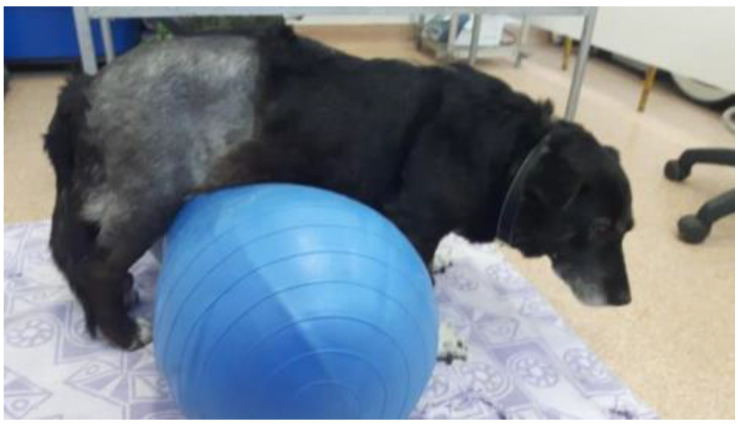
Inducing the standing position with the help of exercise rollers.

**Figure 3 animals-13-01398-f003:**
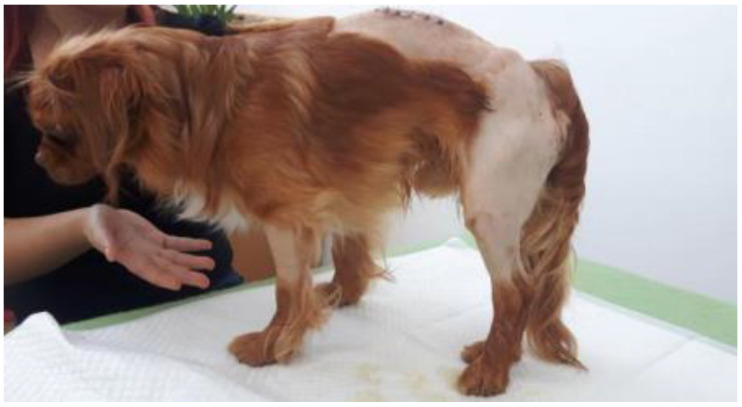
Inducing the standing position independently but with loss of balance.

**Figure 4 animals-13-01398-f004:**
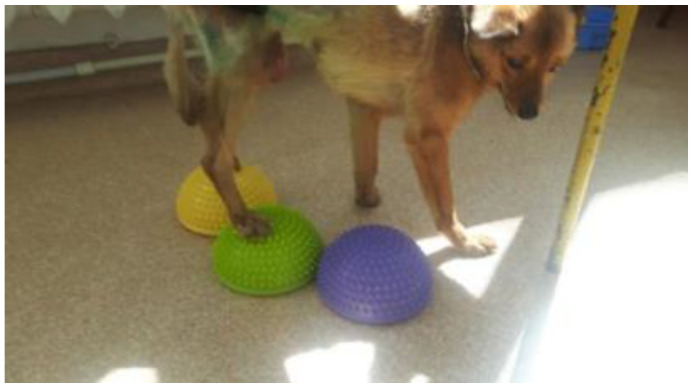
Regaining proprioception and balance.

**Figure 5 animals-13-01398-f005:**
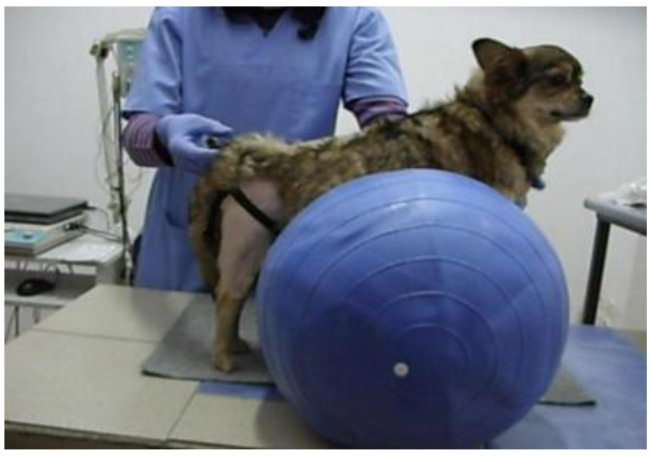
Changing the center of gravity forces the animal to adapt its center of gravity.

**Figure 6 animals-13-01398-f006:**
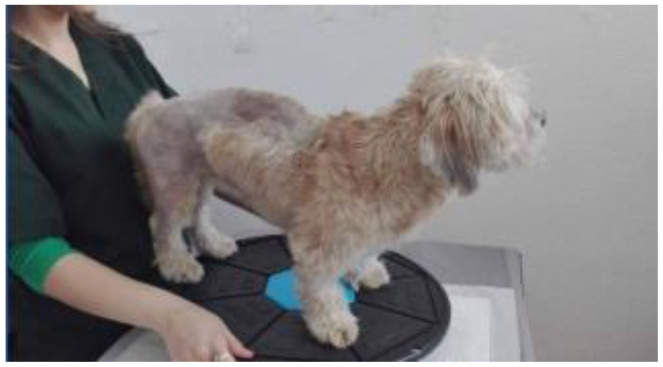
Balancing platform exercises.

**Figure 7 animals-13-01398-f007:**
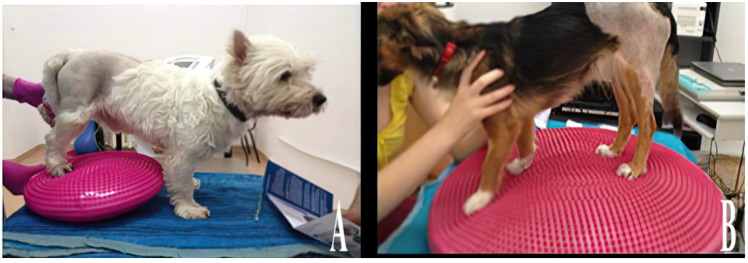
Exercises performed on a small mattress (**A**) and on a large mattress (**B**).

**Figure 8 animals-13-01398-f008:**
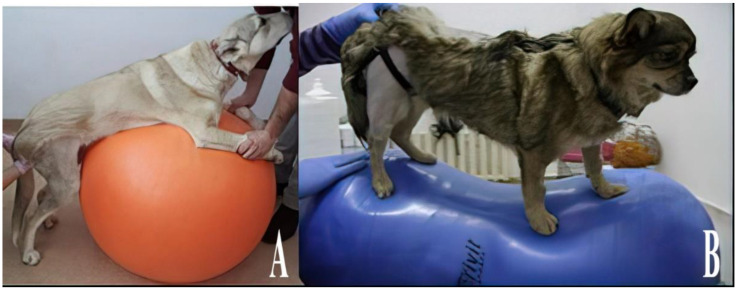
Exercises performed on a physio ball (**A**) and a balance roller (**B**).

**Table 1 animals-13-01398-t001:** Sites and types of lesions (60 patients).

Site of Lesion	Number of Cases	%	IVDE		Trauma	
			Number of Cases	%	Number of Cases	%
Th9–Th10	2	3.33	2	3.33		
Th10–Th11	6	10	4	6.67	2	3.33
Th11–Th12	22	36.67	20	33.33	2	3.33
Th13–L1	29	48.33	26	43.33	3	5.00
L1–L2	1	1.67	1	1.67		
Total	60	100	53	88.33	7	11.67

IVDE—intervertebral disc extrusion; Th—thoracal vertebrae; L—lumbar vertebrae.

**Table 2 animals-13-01398-t002:** The spinal walking and no spinal walking groups (n = 60).

	Spinal Walking Group	No Spinal Walking Group
Dogs	35 (58.33%)	25 (41.67%)
Breeds most represented	Mixed breed (n = 9; 25.71%) Teckel (n = 4; 11.43%) Bichon (n = 5; 14.28%) Pekingese (n = 4; 11.43%) Caniche (n = 2; 5.71%)	Mixed breed (n = 16; 64%) Teckel (n = 4; 16%) Bichon (n = 2; 8%) Caniche (n = 1; 4%) Pug (n = 1; 4%)
Age	m: 54.85 months (range: 3–126)	m: 66.96 months (range: 27–129)
Weight	6.83 kg (range: 1.5–15.7) *	15.59 kg (range: 5.5–45.2)
Dogs with IVDH	32 (91.42%)	21 (84%)
Dogs with traumatic injuries	3 (8.57%)	4 (16%)
Dogs with lesion T9–T10	2 (5.71%)	0
Dogs with lesion T10–T11	2 (5.71%)	4 (40%)
Dogs with lesion T11–T12	7 (20%)	15 (16%)
Dogs with lesion T13–L1	23 (65.71%)	6 (24%)
Dogs with lesion L1–L2	1 (2.85%)	0

* Statistically significant differences between groups *p* = 0.0085.

**Table 3 animals-13-01398-t003:** Degree of recuperation of reflectivity and motility after physiotherapy (60 patients).

No. of Physiotherapy Sessions	Deep Pain and Proprioception	Reflectivity	Motility Recuperation/Description	Gait Score * [37]
Initial (t0)	absent	absent	absent	0
40	absent	absent in 12/60 patients (20%)	47/60 (78.33%) of patients stand up fourfold when drinking water or eating and remain in this position for about 30 s to one minute	5.8 ± 1.44 **
80	absent	absent in 12/60 patients (20%)	47/60 (78.33%) patients stand up quadrupedally and can take at least 10 consecutive steps without falling	8.7 ± 1.31 **
125–320	absent	absent in 12/60 patients (20%)	35/60 (58.33%) developed spinal walking, being able to walk without falling, or fell only sometimes in the case of a quick look, with a lack of coordination between the thoracic and pelvic limbs and difficulties in turning, especially when changing direction, but with recovery of the quadrupedal position in less than 30 s	11.6 ± 1.57 **

* Note: Gait scores were recorded only for patients showing a degree of motility recuperation and who finally developed SW. ** Significant differences from previous evaluation.

## Data Availability

Not applicable.

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
