# Peer review of "Recovery of Spinal Walking in Paraplegic Dogs Using Physiotherapy and Supportive Devices to Maintain the Standing Position"

_animals, 2023, doi:10.3390/ani13081398_

Round 1

Reviewer 1 Report (Previous Reviewer 2)

The manuscript can be accepted in that form. 

Author Response

Thank you very much for your acceptance.

Respectfully,

corresponding author

Reviewer 2 Report (Previous Reviewer 3)

I have reviewed the paper and each and every one of my observations that I describe in red below have already been included.

Simple summary: Generally, the simple summary must be a short paragraph highlighting some of the main points and findings of the study and somehow differs from the abstract. In this case, both are almost the same. Please, revise the Instruction for the authors of the journal for examples of how to write this section.

Line 28: Before describing the animals and parameters included in the present study, please, add the objective of the study in the simple summary and in the abstract (line 42).

Line 32: Since the development of the different devices is the main topic in this study, these devices need to be clearly described.

Line 38: I understand that the length of the simple summary and abstract can limit the amount of information that can be included. However, it is necessary to include a conclusion of your study regarding the devices that were implemented. Which ones had better results or what can the authors conclude about them.

Line 42: One of the key points regarding the application and importance of improving current physiotherapy for dogs with spinal cord injuries is the development of chronic pain. This could be mentioned.

Lines 42-43: Some characteristics of the dogs could be added as well. For example, breed, age, and sex. Also, some information about the study design is missing, such as the length of the physiotherapy and the frequency.

Line 50: Before describing the results, I consider it necessary to add how you evaluated animal recovery was evaluated. Did you use reflex response, gait, muscular tone, and quality of life?

Lines 60-62: Add a reference

Lines 62-64: Please, consider including a brief description of the physiopathology of pain perception, the location of these neurons, and why this is associated with a poor prognosis.

Lines 65-68: I recommend including at least one example of each alternative that is currently applied to dogs suffering spinal cord injury. For example, hemilaminectomy in the case of surgical interventions, TENS/EMS/ultrasound in the case of physiotherapy, and drugs such as meloxicam, firocoxib, cimicoxib as non-steroidal anti-inflammatory drugs, or vitamin E, glucosamine, and chondroitin as nutritional supplements.

Line 72: Include a reference.

Lines 101-103: I recommend including other characteristics of the dogs such as age, sex, breed, and even the time course of the lesion if they were receiving a pharmacological protocol.

Lines 107-108: If you do not want to provide a detailed description of the scoring system, it can be stated that “…according to a scoring system proposed/used/developed by [add reference]”.

Lines 123-125: Although a detailed description of the parameters is included, it might be adequate to state the type of electrostimulation. If it was TENS, MENS, faradic. Mentioning the anatomical site where the electrostimulation was performed could help to improve the methods.

Lines 154-156: Did the authors have a reference to design the devices? Or which physiological/anatomical or medical basis did you use to develop these devices and assure that the material and position of the frames will not be harmful to the animals? If you used previous works as a basis, it would be adequate to state so.   

Line 250: Mind the size difference between “A” and “B” inside the figure.

Discussion: Please, revise the citation style for all the citations (e.g., Galluci et al., Yu et al., Martins et al., Jeong et al., etc.).

Line 366: It would be interesting to discuss and compare the recuperation time in each case since this could help to develop a guide for a dog with a spinal lesion.

This version of the paper is much better than the previous one.

I think it should be accepted, only the authors should adjust the conclusion to the main findings found and it should be more precise.

Author Response

Thank you for thorough analysis and valuable suggestions.

Respectfully,

corresponding author

This manuscript is a resubmission of an earlier submission. The following is a list of the peer review reports and author responses from that submission.

Round 1

Reviewer 1 Report

Thanks for sending this paper again. In the previous review, I suggested to tune down the fact that the dogs were walking because there is no evidence of this, other than the authors saying these dog did walk again with spinal walking. The authors have not made any changes in that sense. It is visible in the text now the dogs do spinal walking but there is no comment to explain that the results are a pure observation. I think that this should be stated.

A few minor comments:

Line 36-37: ‘with a quick look’ is a strange formulation that probably needs to be rephrased; same line 52

Line 42: not motility but mobility; same point 2.9 line 265, table 2, line 414

Two authors have appeared in the list of authors and I am not sure why and this probably needs to be justified.

Thanks

Reviewer 2 Report

The manuscript is not ready to be reviewed. There is a lot of corrections that make text readding difficult. In the supplementary file 2 videos (S3 and S2) are the same and also there is double video S2. It is not complementary with the information from the text and Supplementary Materials (line 460-463).

In the introduction please add more information about possible physiotherapeutic treatment of SCI. What kind of physiotherapy is used, what are the results, is it effective. I do not feel informed well about current knowledge about alternative treatments method for pets diagnosed with SCI. Materials and methods are  chaotically described. Tables in methodology and results are incomplete and repeat. Discussion is good, but the conclusion is very superficial for the researcher and practitioners.

Line 45: I am not sure if the square with details about electrotherapy is essential in the abstract. There is no additional information about other therapies. Please delete it.

Line 101: 60 paraplegic dogs. Please add more details. Mean age, sex, breed or body weight. Table with clinical data would be beneficial.

Line 106: thoracic not thoracal vertebrae. Please correct.

Line 107: Was the clinical and neurological examination performed by the same vet?

Line 121: Please add details about laser therapy and ultrasound therapy. Parameters, how many treatments were performed, where the treatment area was located? Was the treatment area shaved?

Line 122: Please add the producent company, city and country od the Vet Therapy System equipment.

Line 123: Which of the muscles were treated with electrotherapy? Please add more details.

Line 128-129: Please extend the shortcut VMS and Fr. The electrotherapy parameters are not really clear for me. Please explain each parameters step by step. What type of electrodes were used? Was the coat shaved?

Line 161-162: I am not sure if in the materials and methods should be written that some methods “were also very useful”. It is a result! And also described very briefly!

Line 179: “Trolleys and straps are very useful in physiotherapy sessions” it is not a sentence that should be places in methodology.

Line 185-188: Please focus only in the patients included in the study. Delete information about trolley, which is not used in your research!

Line 204-207: Comparison of the used method should be moved to the discussion.

Line 228-229: Describe the details of dynamic balance protocol.

Line 278-280: The same, please do not put any opinion in the methodology. Focus to describe details of the rehabilitation protocol only.  

Line 301: Do not repeat information given in materials and methods (Table 1).

Line 309-313: How you defined spinal walking should be placed in the methodology.

Line 450-451: Lack of the control group is a limitation of yours study and should be written in the discussion. Please think and add more limitations e.g. different dogs breed, different rehabilitation protocol etc.

Line 455-457: Please see comment above.

Reviewer 3 Report

Physiotherapy applied to veterinary medicine is a highly interesting and essential element for neurologic patients. The present article addresses one of the main concerns of guardians and doctors when facing paraplegic patients: the lack of or limited mobility. Developing or testing new devices that can provide support to the animal and improve its rehabilitation. Some aspects could be improved in the manuscript, but the topic is relevant, and the study provides valuable results that can be useful in the clinical daily veterinary practice.

Simple summary: Generally, the simple summary must be a short paragraph highlighting some of the main points and findings of the study and somehow differs from the abstract. In this case, both are almost the same. Please, revise the Instruction for the authors of the journal for examples of how to write this section.

 Line 28: Before describing the animals and parameters included in the present study, please, add the objective of the study in the simple summary and in the abstract (line 42).

Line 32: Since the development of the different devices is the main topic in this study, these devices need to be clearly described.

 Line 38: I understand that the length of the simple summary and abstract can limit the amount of information that can be included. However, it is necessary to include a conclusion of your study regarding the devices that were implemented. Which ones had better results or what can the authors conclude about them.

Line 42: One of the key points regarding the application and importance of improving current physiotherapy for dogs with spinal cord injuries is the development of chronic pain. This could be mentioned.

 Lines 42-43: Some characteristics of the dogs could be added as well. For example, breed, age, and sex. Also, some information about the study design is missing, such as the length of the physiotherapy and the frequency.

Line 50: Before describing the results, I consider it necessary to add how you evaluated animal recovery was evaluated. Did you use reflex response, gait, muscular tone, and quality of life?

 Lines 60-62: Add a reference

 Lines 62-64: Please, consider including a brief description of the physiopathology of pain perception, the location of these neurons, and why this is associated with a poor prognosis.

Lines 65-68: I recommend including at least one example of each alternative that is currently applied to dogs suffering spinal cord injury. For example, hemilaminectomy in the case of surgical interventions, TENS/EMS/ultrasound in the case of physiotherapy, and drugs such as meloxicam, firocoxib, cimicoxib as non-steroidal anti-inflammatory drugs, or vitamin E, glucosamine, and chondroitin as nutritional supplements.

 Line 72: Include a reference.

 Lines 101-103: I recommend including other characteristics of the dogs such as age, sex, breed, and even the time course of the lesion if they were receiving a pharmacological protocol.

Lines 107-108: If you do not want to provide a detailed description of the scoring system, it can be stated that “…according to a scoring system proposed/used/developed by [add reference]”.

Lines 123-125: Although a detailed description of the parameters is included, it might be adequate to state the type of electrostimulation. If it was TENS, MENS, faradic. Mentioning the anatomical site where the electrostimulation was performed could help to improve the methods.

Lines 154-156: Did the authors have a reference to design the devices? Or which physiological/anatomical or medical basis did you use to develop these devices and assure that the material and position of the frames will not be harmful to the animals? If you used previous works as a basis, it would be adequate to state so.   

Line 250: Mind the size difference between “A” and “B” inside the figure.

Discussion: Please, revise the citation style for all the citations (e.g., Galluci et al., Yu et al., Martins et al., Jeong et al., etc.).

 Line 366: It would be interesting to discuss and compare the recuperation time in each case since this could help to develop a guide for a dog with a spinal lesion.